# HER2 Expression Is Predictive of Survival in Cetuximab Treated Patients with *RAS* Wild Type Metastatic Colorectal Cancer

**DOI:** 10.3390/cancers13040638

**Published:** 2021-02-05

**Authors:** Said A. Khelwatty, Soozana Puvanenthiran, Sharadah Essapen, Izhar Bagwan, Alan M. Seddon, Helmout Modjtahedi

**Affiliations:** 1School of Life Science, Pharmacy and Chemistry, Kingston University London, London KT1 2EE, UK; S.Khelwatty@Kingston.ac.uk (S.A.K.); fsr.puvanenthiran@gmail.com (S.P.); S.Essapen@nhs.net (S.E.); Alan.Seddon@Kingston.ac.uk (A.M.S.); 2St Luke’s Cancer Centre, Royal Surrey County Hospital, Guildford GU2 7XX, UK; 3Department of Histopathology, Royal Surrey County Hospital, Guildford GU2 7XX, UK; Izhar.Bagwan@nhs.net

**Keywords:** metastatic colorectal cancer, HER2, cetuximab, predictive biomarker

## Abstract

**Simple Summary:**

Treatment with anti-epidermal growth factor receptor (EGFR) monoclonal antibodies (mAbs)—cetuximab and panitumumab—produced clinical benefits in a subset of patients with metastatic colorectal cancer (mCRC). Here, the authors investigated the relative expression and predictive value of HER family members in 144 patients with CRC. They found high levels of expression of HER2 in patients treated with cetuximab; these were associated with shorter progression free survival (PFS). The results provide support for the emergence of HER2 as a therapeutic target in patients with mCRC.

**Abstract:**

The overexpressed HER2 is an important target for treatment with monoclonal antibody (mAb) trastuzumab, only in patients with breast and gastric cancers, and is an emerging therapeutic biomarker in metastatic colorectal cancer (mCRC) treated with anti-epidermal growth factor receptor (EGFR) mAbs cetuximab and panitumumab. In this study, we investigated the relative expression and predictive value of all human epidermal growth factor receptor (HER) family members in 144 cetuximab-treated patients with wild type *RAS* mCRC. The relative expression of EGFR and HER2 have also been examined in 21-paired primary tumours and their metastatic sites by immunohistochemistry. Of the 144 cases examined, 25%, 97%, 79%, 48%, and 10% were positive for EGFR, HER2, HER3, and HER4 and all four HER family members, respectively. The expression of EGFR was an indicator of poorer overall survival and the membranous expression of HER2 and HER3 3+ intensity was associated with a shorter progression free survival (PFS). In contrast, the cytoplasmic expression of HER2 was associated with better PFS. In 48% and 71% of the cases, there were discordance in the expression of EGFR or one or more HER family members in paired primary and related metastatic tumours, respectively. Our results implicate the importance of a large prospective investigation of the expression level and predictive value of not only the therapeutic target (i.e., EGFR protein) but also HER2 and other HER family members as therapeutic targets, or for response to therapy with anti-EGFR mAbs and other forms of HER inhibitors, in both the primary tumours and metastatic sites in mCRC.

## 1. Introduction

Colorectal cancer (CRC) is a heterogenous disease and one of the leading causes of cancer deaths worldwide [1]. In the USA, it is predicted to be the third most commonly diagnosed cancer (147,950) and the second leading cause of cancer deaths (53,200) [1,2]. Identification of cell surface antigens with high levels of expression in colorectal cancer resulted in the development and approval of several monoclonal antibody-based drugs for use in targeted therapy of metastatic colorectal cancer (mCRC) [3,4]. Currently, anti-epidermal growth factor receptor (EGFR) monoclonal antibodies (mAbs) cetuximab and panitumumab, anti-vascular endothelial growth factor (VEGF) mAb bevacizumab, and anti-VEGFR2 mAb ramucirumab are approved for the treatment of patients with metastatic colorectal cancer [5,6,7,8]. Additionally, three checkpoint inhibitors namely anti-PD-1 mAb nivolumab alone or in combination with anti-cytotoxic T-lymphocyte-associated protein 4(CTLA-4) mAb ipilimumab and anti-programmed cell death protein 1 (PD-1) pembrolizumab have been approved for the treatment of patients with metastatic microsatellite instability-high (MSI-H) or mismatch repair deficient (dMMR) colorectal cancer [9,10]. However, therapeutic benefits are seen in a subset of such patients and the duration of response can also be short. Indeed, there is currently no reliable predictive biomarker for the selection of patients who benefit from therapy with anti-EGFR antibodies [4]. In some studies, tumour heterogeneity, mutations in *RAS* and the mAb binding sites have been associated with resistance to anti-EGFR mAbs in patients with mCRC [11,12]. There is a need for the identification of more reliable biomarkers for the response to therapy with anti-EGFR antibodies in patients with mCRC [4,13]. 

EGFR (HER1/ErbB1) is a member of the type I growth factor receptor family of tyrosine kinases (also known as ErbB/HER family), which consists of three other members namely HER2 (neu, ErbB2), HER3 (ErbB3), and HER4 (ErbB4) [4]. EGFR is an important therapeutic target for therapy with anti-EGFR antibodies, however, interestingly, the expression of EGFR is not used for patient selection due to reports of response to cetuximab in mCRC patients with EGFR negative tumours [14]. Indeed, while *RAS* mutation is currently an important negative predictive biomarker for the response [15], not all patients with a wild type *KRAS* and *NRAS* respond to, or gain benefit from therapy with anti-EGFR mAbs. On the contrary, considerable objective responses have been reported in mCRC patients with *KRAS* mutations treated with cetuximab [16,17,18]. More recently, other studies have also reported emergence of *KRAS* mutations in liver metastases after anti-EGFR treatment, highlighting the role of intra-tumour heterogeneity as a major contributing factor for intrinsic and/or acquired resistance to anti-EGFR mAbs [19]. 

In light of the heterogenous nature of tumours and the high degree of crosstalk between different receptor signalling pathways, in a limited number of studies, we and others have reported the co-expression and prognostic significance of all members of the HER family members and significant alterations were found in the expressions of HER1-4 in primary and corresponding metastatic lesions in patients with mCRC [20,21,22,23]. 

In the present study, we report for the first time the co-expression and predictive value of all HER family members in 144 mCRC patients with *RAS* wild type, as well as changes in such expression before and post cetuximab treatment, and their association with survival and response to therapy. 

## 2. Materials and Methods

In this retrospective study, 144 mCRC surgically resected and biopsy samples as well as 21 paired metastatic specimens from patients between 2010 and 2018, treated with FOLFOX (oxaliplatin and modified de Gramont) or FOLFIRI (irinotecan and modified de Gramont) plus cetuximab were used. Before examination, ethical approval was obtained via Research Ethics Committee (REC) of the Health Research Authority (HRA) (Integrated Research Application System Project ID: 228447) and from the Research and Development Committee of the Royal Surrey County Hospital. Response to cetuximab therapy was evaluated using official reports, radiographic studies, including follow up computed tomography scans and the Response evaluation criteria in solid tumors (RECIST) criteria v1.1 [24]. For data analysis, response to cetuximab therapy was grouped as response (full response or partial response) and no response (stable disease or disease progression), clinical endpoints of death and recurrence were examined for overall survival (OS) and progression free survival (PFS) to investigate the prognostic and predictive value of the biomarkers in line with the reporting recommendations for tumour marker prognostic studies (REMARK) guidelines [25].

Tumour samples were routinely determined for *RAS* variants, prior to treatment at Surrey Pathology Services, Royal Surrey County Hospital, Guildford (UK), using an in-house developed panel of amplicons, which generates a library using the Thermo Fisher Ion Chef. The library was then sequenced using the Thermo Fisher S5 platform and the Ion Torrent server and the data analysed using the Thermo proprietary bioinformatics pipeline. 

### 2.1. Immunohistochemistry

Immunohistochemical (IHC) staining was carried out using the Ventana Discovery ULTRA IHC/ISH System (Roche, UK), as described previously [26]. The following primary monoclonal antibodies, mouse anti-wild type EGFR (wtEGFR) specific, does not cross react with the truncated EGFRvIII (M7298) 1:50, (Agilent, Stockport, UK), HER2 (3B5) 1:200, (Insight Biotechnology, London, UK), HER3 (SP71) 1:50, (Abcam, Cambridge, UK), HER4 (HFR1) 1:100, (Insight Biotechnology, London, UK). 

### 2.2. Scoring Criteria

The immunostaining of the tumour sections was scored as described previously [26]. Briefly, the HER immunostaining of the tumour sections was scored based on cut-off values of >5%, >10%, >20%, and >50% of tumour cells, intensities of negative 0, weak 1+, moderate 2+, and strong 3+ and localization of the HER immunostaining (i.e., membrane, cytoplasm or nucleus of the cells) Appendix A. Scoring was conducted by two independent observers (including a consultant histopathologist) who were blinded to all clinical information; any disparity in scoring was resolved by simultaneous reassessment of the staining by both observers. 

### 2.3. Statistical Analysis

Statistical analysis was carried out in SPSS statistics 26 (SPSS Inc. Armonk, NY, USA) and, where applicable, Fishers exact test, Kaplan–Meier survival plots, log rank-test, and Cox survival regression models were used. All clinicopathological parameters (Table 1) were initially considered for inclusion in Cox survival regression models. Additionally, OS and PFS analysis were conducted by omitting the missing data. *p* ≤ 0.05 was considered statistically significant. 

## 3. Results

### 3.1. Clinicopathological Features

The median patient follow-up time was 4 years, median OS was 28.5 months, and median PFS was 19 months. All patients received FOLFIRI (irinotecan and modified de Gramont), plus cetuximab or FOLFOX (oxaliplatin and modified de Gramont), plus cetuximab therapies as first line chemotherapy. Patients with G1 and G2 tumours were found to have a better OS (*p* = 0.022), while patients treated with FOLFIRI plus cetuximab therapy exhibited longer PFS (*p* = 0.007). A summary of patient clinicopathological characteristics is shown in Table 1.

### 3.2. Immunohistochemical Expression of HER Family Members in Primary Tumours

For the first time in this study, we determined the expression of all four HER family members as well as their co-expression, in tumours with wild type *RAS* from mCRC patients before and after treatment, at different cut-off values and staining intensity, and the results are presented in Table 2. The anti-EGFR mAb used in this study is specific for the wild type EGFR, and does not cross react with the truncated EGFRvIII and, therefore, detects only the level of ligand-dependent EGFR. For example, at cut off value of >5%, of the 144 cases examined, 25%, 97%, 79%, and 48% were found to be positive for wtEGFR, HER2, HER3, and HER4 respectively. wtEGFR and HER3 predominantly stained the membrane, while HER2 exhibited cytoplasmic and HER4 nuclear staining patterns in the majority of the cases (Figure 1, Table 2). 

The EGFR, HER2, HER3 staining intensity of 3+ were present in 1%, 6%, and 7% of the cases examined respectively. None of the tumours had HER4 staining intensity of 3+ (Table 2, Figure 1). 

The co-expression of the HER2 and HER3 was noted in the majority of the cases (76%), while 10% co-expressed all four HER family members (Table 2). It is also noteworthy that, in this study, we did not find the co-expression of HER family members to have a significant statistical difference in both OS and PFS survival analyses (Appendix A). 

### 3.3. HER Expression in Primary CRC and Their Corresponding Metastases 

Of the 21 patients with primary and paired metastasis (14 metachronous and 7 synchronous metastases), 71% (15/21 patients) exhibited an overall change in one or more HER expression (Table 3). Interestingly, in 38% (8/21) of cases, examined both the primary tumour and related metastasis were EGFR negative (staining present in less than 5% of tumour cells or staining intensity of 0) and 48% (10/21) patients had discordance in the expression of wtEGFR in the primary tumours and related metastasis. Of these six EGFR positive primary tumours, the corresponding metastatic sites were EGFR negative, whereas in four EGFR positive metastasis, the primary tumours were EGFR negative (Table 3). Similarly, there were 19%, 14%, and 52% discordance in the expression of HER2 (4/21), HER3 (3/21), and HER4 (11/21) in the primary tumours and related metastatic sites (Table 3).

### 3.4. wtEGFR Expression is Significantly Associated with Adverse Overall Survival

Using the wild type specific anti-EGFR mAb, the expression of EGFR and its association with OS was investigated using Kaplan–Meier curves and log rank-test. We found that OS was significantly poorer in patients with cytoplasmic expression of wtEGFR (*p* = 0.003) in this study (Figure 2A). The expression of wtEGFR was an indicator of poorer OS (hazard ratio (HR) 3.584, CI 1.455–8.826, *p* = 0.006) and remained an independent prognostic biomarker of worse OS (*p* = 0.007) (Table 4). Interestingly, we also found the expression of wtEGFR at cut-off values of >20% and >50% to be significantly associated with poorer OS (*p* = 0.008) (Figure 2A), and an independent biomarker for worse OS (HR 2.914, CI 1.064–7.896, *p* = 0.038, HR 4.810, CI 1.320–17.524, *p* = 0.017 respectively) (Table 4).

### 3.5. HER2 and HER3 Expression Impacts Progression Free Survival 

A significant association was found between the HER2 expression and progression free survival (Figure 2B). Interestingly, the membranous expression of HER2 was associated with a shorter progression free survival (*p* = 0.004), while the cytoplasmic expression of HER2 was found to be predictive of a better PFS (*p* = 0.012) (Figure 2B). 

Univariate analysis further showed that HER2 membranous expression increased the risk of shorter PFS by 2-fold (HR 2.097, CI 1.242–3.542, *p* = 0.006). It also remained an independent predictive biomarker of shorter PFS when analysed in a multivariate model (*p* = 0.007). Equally, cytoplasmic HER2 remained an independent predictive biomarker of better PFS when analysed in univariate (HR 0.518, CI 0.305–0.879, *p* = 0.015) and multivariate (*p* = 0.004) regression models (Table 4). 

As shown in Table 2, HER3 was the second most commonly expressed receptor in this study. Using Kaplan–Meier survival curves we found HER3 expression with a score of 3+ intensity to be significantly associated with shorter PFS (Figure 2, *p* = 0.045). While this association was found to be significant, when analysed in the univariate and multivariate cox-survival regression model, it did not reach significance to be an independent biomarker of PFS when analysed (Table 4).

## 4. Discussion

The approval of the anti-EGFR mAbs—cetuximab and panitumumab—for the treatment of mCRC and, in recent years, necitumumab for lung cancer [5,7,27,28,29], emphasises the importance of EGFR as a therapeutic target in human cancers. Despite a response of short duration in patients with mCRC, the inclusion of anti-EGFR therapy to chemotherapy with FOLFOX or FOLFIRI has been shown to increase response rates and convert patients with inoperable liver disease to potentially resectable metastatic disease [30]. However, owing to various reasons, such as difference in the scoring criteria, there are no clear associations between the expression of EGFR protein and response to therapy [4]. Therefore, it is currently recommended that patient selection should be based on the wild type (WT) molecular status of *RAS* mutations rather than any expression of the EGFR protein, which is the actual target for anti-EGFR antibodies [15,31]. 

Indeed, a well-characterized negative predictor of response to antibody therapy in mCRC is mutated *RAS* [32]. However, objective response rates in patients with mutated *KRAS* have been observed in several studies, while others have reported response rates of up to 40% only in patients without mutations in *KRAS* or *BRAF* [18]. In addition, others have reported emergence of *KRAS* mutations in liver metastases not detectable in the colonic biopsy, highlighting the role of intra-tumour heterogeneity as a major contributing factor for intrinsic and/or acquired resistance [19]. 

Therefore, it is evident that it is vitally important to identify a more reliable predictive biomarker of response to anti-EGFR therapy and/or more robust therapeutic targets for monoclonal antibody or small molecule tyrosine kinase inhibitor-based therapy in CRC [3,4]. Consequently, in the present study we aimed to investigate the predictive value of the wtEGFR by determining its expression level, before and after anti-EGFR therapy in patients with mCRC. In addition, for the first time, in this study we determined the expression level of all other HER family members (HER2, HER3, and HER4), which act as EGFR heterodimerisation partners, in mCRC patients with wild type *RAS* treated with standard chemotherapy in combination with cetuximab.

In the present study, we report the immunohistochemical expression of the receptors using various cut-off values for the percentage of staining as well as the location and intensity of the immunostaining to maintain consistency with our previous studies [33] (Table 2). We found the overall expression of wtEGFR to be a negative prognostic marker of overall survival as both cytoplasmic expression of wtEGFR as well as positivity at more than 50% cut-off value were found to be significantly associated with a shorter overall survival in this study. Due to the fact that the anti-EGFR mAb cetuximab is incapable of binding to the cytoplasmic wtEGFR, to induce the antibody-dependent cell-mediated cytotoxicity (ADCC), this association was largely expected [34]. 

Interestingly, while the predominant pattern of the EGFR staining was membranous, these findings are in contrast with our previous study in which the expression of the EGFR was mostly present in the cytoplasm where we found a positive relationship between this expression and response to treatment with cetuximab [26]. A possible explanation for this discordance could be due to the fact that the present study included a significantly larger proportion of patient tumours that were naïve to cetuximab treatment when the EGFR expression was determined. Indeed, it is a well-investigated mechanism of action of cetuximab, following binding with the EGFR, to downregulate the receptor by inducing receptor internalisation [35,36,37]. Interestingly, while overexpression of membranous EGFR is of great importance for targeted therapy with antibodies, in this study the EGFR staining intensity of 3+ was only present in 1% of the cases examined (Table 3). 

The abundant expression of HER2 (97%) and its significance as a predictive biomarker of PFS, was an interesting and important observation of the present study. We found the membranous expression of HER2 to be significantly associated with a shorter PFS, while the opposite was found to be true for the cytoplasmic expression of HER2 in this study. Several studies have examined the prognostic role of HER2 expression in colorectal cancer [38,39,40,41,42], but very few have investigated the relative expression and predictive value of HER2 and other HER family receptors in mCRC. We found a high expression of HER3 to be associated with a shorter PFS in this study and 26% and 7% of the cases examined had staining intensity of 2+ and 3+, respectively (Table 3). These findings support the need for more detailed investigation of HER2 and HER3, which can form heterodimers with the EGFR, as predictive biomarkers for the response to therapy and therapeutic targets for treatment with anti-EGFR antibodies and other forms of HER inhibitors, when used alone and or in combinations in mCRC [37,43]. 

HER2 is known to preferentially dimerise with HER3, to initiate downstream signalling pathways for cell proliferation as well as angiogenesis [44]; therefore, both HER2 and HER3 receptors could represent effective therapeutic targets. Indeed, these receptors are not only targeted for by mAb therapy, such as anti-HER2 antibody trastuzumab, which has already been approved for metastatic gastric cancer and breast cancer, anti-HER3 antibody MM-121 and Pan-HER, an antibody mixture targeting EGFR, HER2, and HER3, which is in early clinical development, but also using small molecule tyrosine kinase inhibitors, such as lapatinib, and pan-HER blockers, such as afatinib, in various patient populations, including CRC patients [43,45,46,47]. Interestingly, in a recent phase II, multicentre, open-label study treatment with trastuzumab conjugated with deruxtecan (T-DXd) have shown remarkable activity in patients with HER2-expressing mCRC refractory to standard therapies [48]. In our study, we found that 10% of cases had HER2 staining present in >10% tumour cells and 6% with staining intensity of 3+. These observations, along with the strong associations between HER2 and HER3 expressions and PFS found in the present study, provide further support that HER2 and HER3 measurements in tumours from mCRC patients may be useful as predictive biomarkers for the response to therapy. Furthermore, these measurements may provide vital information to determine additional targets for therapeutic interventions in patients with colorectal cancer and warrant further investigation.

Unlike the other HER family members, there are very few studies that have investigated the role of HER4 in human cancers, in particular mCRC [20,21,49,50]. To our knowledge, this is the first study to report a significant number of mCRC patients treated with cetuximab expressing HER4. While we did not find any significant association between HER4 expression and OS or PFS in this study, others have reported HER4 as an independent prognostic factor. Biaocchi and colleagues investigated the expression of HER4 in 109 CRC patients with high-risk of recurrence after radical surgery and found positive membranous expression of HER4 to be an independent prognostic factor for recurrence [20]. In a more recent study, HER4 expression was found to promote the progression of colorectal cancer through epithelial-mesenchymal transition and was related to an unfavourable clinical outcome in patients with CRC [51]. Despite the membranous cellular localisation of HER4 in our study, which was around 3%, we found overall 48% of the cases were HER4 positive. The relatively common expression of HER4 in the present study, which can act as an EGFR heterodimerisation partner, supports the need for further investigation of its predictive value for the response to therapy with anti-EGFR mAbs in patients with mCRC. Currently, there are a number of ongoing clinical trials further assessing the efficacy of HER family inhibitors in mCRC (EudraCT Number: 2020-000540-60, 2012-002128-33, 2017-003466-28, 2020-001574-29, 2013-002872-42, 2014-003277-42; NCT03043313, NCT03457896, NCT04603287). 

Finally, in our study we found an overall response rate of 65% to chemotherapy plus cetuximab in patients with wild type *RAS* status, which means that 35% of the patients with no detectable mutations on *RAS*/*RAF* still exhibit an intrinsic resistance to therapy. In our view, one of the biggest issues is the fact that often a small colorectal biopsy is used for the single assessment of the patient’s cancer *RAS* status, which can often pre-date the metastases significantly. Indeed, as shown by Baretti and colleagues, the reported emergence of *KRAS* mutations in liver metastases may occur after anti-EGFR treatment and not be detectable in the colonic biopsy [19]; however, more relevant, is the expression level of the therapeutic target, which is often overlooked and/or not determined prior to commencement of the treatment. Interestingly, we have shown that up to 71% of the primary and paired metastasis exhibited an overall change in one or more HER expression in this study. Similarly, other studies reported that EGFR expression was lost in 33% of metastasising primary colorectal cancer tumours [23]. 

## 5. Conclusions

Taken together, we believe that for targeted therapy of cancer, the expression level of the target antigen, and its status must not be overlooked, and it is time to get back to the target, which in the case anti-EGFR antibodies are the EGFR protein and its dimerisation partners [52,53,54]. We, therefore, recommend the need for more detailed examination of both the primary tumours and corresponding metastasis to accurately determine the level, cellular location, and intensity of the therapeutic target, which is the EGFR protein as well as *RAS* status, when considering the use of EGFR targeted therapies in mCRC. Furthermore, such studies may also help in the identification of those patients who may gain additional benefit from the targeted therapy with the other forms of HER inhibitors. 

## Figures and Tables

**Figure 1 cancers-13-00638-f001:**
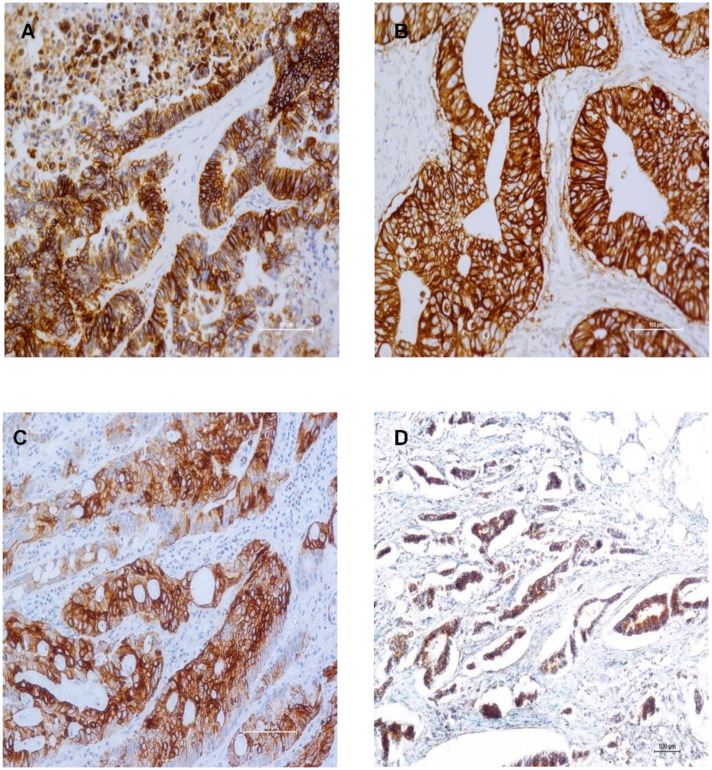
Immunostaining of Human Epidermal Growth Factor Receptor family members in mCRC specimens. Immunostaining of wild type EGFR (wtEGFR) (**A**), HER2 (**B**), HER3 (**C**), and HER4 (**D**) in formalin fixed paraffin embedded tumour sections stained immunohistochemically using the Ultra Discovery Autostainer as described previously.

**Figure 2 cancers-13-00638-f002:**
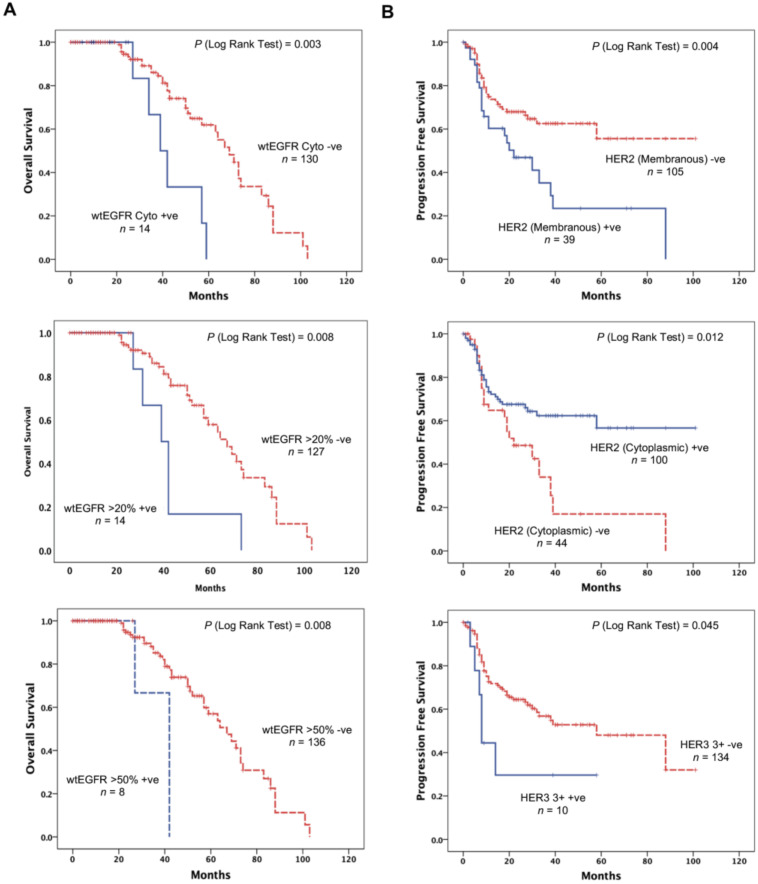
The association between HER family members and OS in mCRC patients treated with cetuximab. Kaplan–Meier survival curves showing the impact on the OS of the patients with localised cytoplasmic wtEGFR expression, wtEGFR expression at >50% cut-off value, and HER4 expression at >20% cut off value (**A**), and impact on the PFS of patients with localised membranous and cytoplasmic HER2 expression, HER3 3+ staining intensity (**B**). A log-rank test value of *p* ≤ 0.05 was considered statistical significance.

**Table 1 cancers-13-00638-t001:** Clinicopathological parameters and survival of 144 metastatic colorectal cancer (mCRC) patients treated with anti-epidermal growth factor receptor (EGFR) (mAb) cetuximab or panitumumab. Progression free survival (PFS) and overall survival (OS) relative to the indicated features were determined by Kaplan–Meier analysis and the log-rank test. *p*-Value of ≤0.05 was considered significant.

Characteristics	Number of Patients (%)	PFS in Months (Mean ± SE)	95% CI	*p*-Value	OS in Months (Mean ± SE)	95% CI	*p*-Value
Age in years				*NS*			*NS*
≤70	85 (59)	53.7 ± 5.8	42.3–65.1		61.5 ± 4.6	52.5–70.6	
>70	59 (41)	55.5 ± 5.6	44.4–66.5		67.6 ± 4.7	58.3–76.9	
Gender				*NS*			*NS*
Male	101 (70)	47.1 ± 4.4	38.5–55.6		63.8 ± 3.2	57.4–70.1	
Female	43 (30)	71.9 ± 7.0	58.1–85.6		59.4 ± 6.6	46.4–72.4	
Tumour Type				*NS*			*NS*
Resection	76 (53)	52.9 ± 6.0	41.2–64.6		63.2 ± 4.1	55.1–71.3	
Liver Metastasis	11 (8)	39.14 ± 8.7	22.4–56.1		64.1 ± 5.5	53.3–74.9	
Biopsy	57 (39)	48.4 ± 4.8	39.0–57.8		61.4 ± 6.5	48.5–74.2	
T stage *				*NS*			*NS*
<T4	46 (32)	51.2 ± 6.9	37.5–64.9		63.8 ± 4.9	54.2–73.4	
T4	33 (23)	38.8 ± 4.2	30.6–47.0		61.7 ± 6.6	48.9–74.6	
N Stage *				*NS*			*NS*
<N2	41 (29)	50.5 ± 6.4	38.1–62.9		63.9 ± 4.3	55.6–72.3	
N2	38 (26)	63.3 ± 7.8	48.0–78.5		65.1 ± 11.3	43.0–87.2	
M Stage *				*NS*			*NS*
Mx/M0	70 (49)	53.6 ± 6.1	41.5–65.6		64.7 ± 4.1	56.6–72.8	
M1	11 (8)	25.8 ± 4.1	17.8–33.8		34.8 ± 3.0	28.9–40.7	
Vascular Invasion *				*NS*			*NS*
V0	30 (21)	49.7 ± 7.6	34.8–64.7		63.6 ± 5.7	52.3–74.8	
V1	49 (34)	54.7 ± 7.1	40.8–68.7		65.5 ± 5.9	53.9–77.0	
LVI *				*NS*			*NS*
Yes	39 (27)	55.5 ± 7.8	40.2–70.7		64.6 ± 4.9	54.8–74.4	
No	39 (27)	50.1 ± 6.7	37.1–63.2		63.6 ± 6.4	51.1–76.1	
Grade *				0.022			*NS*
G1&G2	78 (54)	61.3 ± 6.1	49.4–73.1		63.0 ± 4.1	54.9–71.1	
G3	32 (22)	31.9 ± 6.4	19.4–44.6		57.7 ± 6.2	45.4–69.9	
Apical Node *				*NS*			*NS*
Negative	67 (47)	53.4 ± 6.2	41.2–65.5		65.5 ± 4.1	56.4–72.7	
Positive	12 (8)	30.4 ± 4.4	21.9–38.9		36.7 ± 2.6	31.6–41.7	
Chemotherapy				*NS*			0.007
FOLFOX + cetuximab	25 (17)	45.8 ± 5.9	34.2–57.6		47.6 ± 6.5	34.8–60.5	
FOLFIRI + cetuximAb	78 (54)	36.4 ± 3.7	29.1–43.6		65.5 ± 3.6	58.5–72.5	

* data for T stage, N stage, Vascular invasion, and grade missing in some cases due to being biopsy and/or liver metastases patients. OS and PFS analysis were conducted by omitting the missing data.

**Table 2 cancers-13-00638-t002:** Immunohistochemical expression of HER family members and their co-expressions in 144 mCRC patients treated with anti-EGFR cetuximab.

Variables		No. of Positive Tumours (%)									
		% Positive Tumour Cells				Intensity			Location		
		>5	>10	>20	>50	1+	2+	3+	Mem	Cyto	Nuc
EGFR		36	28	17	8	26	15	2	31	14	-
		(25)	(19)	(12)	(6)	(18)	(10)	(1)	(22)	(10)	
HER2		139	125	103	89	90	37	8	39	100	-
		(97)	(87)	(72)	(62)	(63)	(26)	(6)	(27)	(69)	
HER3		114	79	55	29	67	38	10	70	43	-
		(79)	(55)	(38)	(20)	(47)	(26)	(7)	(49)	(30)	
HER4		69	59	34	14	67	2	0	4	27	38
		(48)	(41)	(24)	(10)	(47)	(1)	(0)	(3)	(19)	(26)
**Co-expression of HER family (%)**											
**EGFR**	**EGFR**	**EGFR**	**HER2**	**HER2**	**HER3**	**EGFR**	**EGFR**	**EGFR**	**HER2**	**EGFRHER2**	
**HER2**	**HER3**	**HER4**	**HER3**	**HER4**	**HER4**	**HER2HER3**	**HER2HER4**	**HER3HER4**	**HER3HER4**	**HER3HER4**	
34	29	17	109	68	57	27	15	18	56	15	
(24)	(20)	(12)	(76)	(47)	(40)	(19)	(10)	(13)	(39)	(10)	

Mem, Membranous; Cyto, Cytoplasmic; Nuc, Nuclear.

**Table 3 cancers-13-00638-t003:** Summary of HER expression in primary and corresponding metastatic lesions in CRC patients treated with anti-EGFR cetuximab.

Case	Sex	Age	Type of Metastasis	wtEGFR 1° Met		HER2 1° Met		HER3 1° Met		HER4 1° Met		Overall Change in Expression	Response
1	Female	64	Metachronous	−ve	−ve	−ve	−ve	+ve	+ve	−ve	+ve	Yes	Yes
2	Female	52	Metachronous	+ve	−ve	−ve	−ve	+ve	−ve	−ve	+ve	Yes	Yes
3	Male	52	Synchronous	−ve	−ve	−ve	−ve	+ve	+ve	+ve	+ve	No	Yes
4	Male	60	Synchronous	+ve	+ve	+ve	−ve	+ve	+ve	+ve	−ve	Yes	No
5	Male	88	Metachronous	+ve	−ve	−ve	−ve	+ve	+ve	+ve	−ve	Yes	No
6	Male	61	Metachronous	−ve	−ve	−ve	−ve	+ve	+ve	+ve	−ve	Yes	Yes
7	Female	49	Synchronous	−ve	+ve	−ve	+ve	+ve	+ve	+ve	+ve	Yes	No
8	Female	69	Metachronous	−ve	−ve	−ve	−ve	+ve	+ve	+ve	+ve	No	No
9	Male	65	Metachronous	−ve	+ve	−ve	−ve	+ve	+ve	+ve	−ve	Yes	No
10	Female	50	Metachronous	−ve	−ve	−ve	−ve	+ve	+ve	+ve	+ve	No	No
11	Male	56	Metachronous	+ve	−ve	−ve	−ve	+ve	+ve	+ve	−ve	Yes	Yes
12	Male	70	Synchronous	+ve	+ve	−ve	−ve	+ve	+ve	+ve	+ve	No	Yes
13	Female	76	Metachronous	−ve	−ve	−ve	−ve	+ve	+ve	+ve	+ve	No	Yes
14	Male	62	Synchronous	+ve	−ve	+ve	−ve	+ve	+ve	−ve	+ve	Yes	Yes
15	Female	58	Metachronous	−ve	+ve	−ve	+ve	+ve	+ve	−ve	+ve	Yes	Yes
16	Male	65	Synchronous	+ve	+ve	−ve	−ve	+ve	+ve	+ve	+ve	No	Yes
17	Male	69	Metachronous	+ve	−ve	−ve	−ve	+ve	+ve	+ve	−ve	Yes	Yes
18	Male	74	Metachronous	−ve	−ve	−ve	−ve	+ve	+ve	−ve	+ve	Yes	Yes
19	Male	69	Metachronous	+ve	−ve	−ve	−ve	−ve	+ve	+ve	+ve	Yes	Yes
20	Male	35	Synchronous	−ve	−ve	−ve	−ve	+ve	−ve	+ve	+ve	Yes	No
21	Male	34	Metachronous	−ve	+ve	−ve	−ve	+ve	+ve	+ve	+ve	Yes	Yes

**Table 4 cancers-13-00638-t004:** Univariate and multivariate analysis related to PFS and OS in 144 mCRC treated with anti-EGFR mAb cetuximab. *p*-Value of ≤0.05 was considered significant.

Variables	Progression Free Survival (PFS)					
	Univariate			Multivariate		
	HR	95% CI	*p*-Value	HR	95% CI	*p*-Value
HER2 (Membranous)	2.097	1.242–3.542	0.006	2.560	1.295–5.059	0.007
HER2 (Cytoplasmic)	0.518	0.305–0.879	0.015	0.367	0.185–0.728	0.004
HER3 (3+)	2.307	0.986–5.398	0.054	-	-	NS
	**Overall Survival (OS)**					
	**Univariate**			**Multivariate**		
	**HR**	**95% CI**	***p*-Value**	**HR**	**95% CI**	***p*-Value**
wtEGFR (Cytoplasmic)	3.584	1.455–8.826	0.006	3.822	1.446–10.103	0.007
wtEGFR (>20%)	3.084	1.274–7.464	0.013	2.914	1.064–7.986	0.038
wtEGFR (>50%)	4.473	1.309–15.287	0.017	4.810	1.320–17.524	0.017
HER2	0.213	0.62–0.734	0.014	-	-	NS

HR, hazard ratio; CI, confidence interval; NS, Not significant.

## Data Availability

The data that support the findings of this study are available by reasonable request from the corresponding author. The data are not publicly available due to privacy or ethical restrictions.

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
