# Peer review of "HER2 Expression Is Predictive of Survival in Cetuximab Treated Patients with *RAS* Wild Type Metastatic Colorectal Cancer"

_cancers, 2021, doi:10.3390/cancers13040638_

Round 1
Reviewer 1 Report
Khelwatty and colleagues presented an interesting research article aimed at elucidating the expression levels and the prognostic significance of the members of the HER family in a wide cohort of metastatic colorectal cancer patients treated with EGFR monoclonal antibodies. Through immunohistochemistry evaluations, the authors analyzed the expression levels of EGFR, HER2, HER3 and HER4 correlating these expression levels with the overall survival and progression-free survival of patients. Overall, the manuscript is well-written and the study interesting. Although the research design is simple, the findings of the authors could pave the way to novel clinical trials aimed at assessing the therapeutic effects of mAbs against HER2 or other receptors in metastatic colorectal cancer patients over-expressing these molecules. Below are reported some minor/major comments that will improve the quality of the manuscript:
1) As the authors use the name of the active principles cetuximab and panitumumab in the Abstract section, they should be consistent using the name “trastuzumab” instead of the commercial one “Herceptin”;
2) In line 40-42, please avoid the repetition of “in the USA”;
3) Please, revise the following sentence: “Currently, the anti-epidermal growth factor receptor (EGFR) monoclonal antibodies (mAbs) cetuximab and panitumumab and 45 anti-vascular endothelial growth factor (VEGF) mAb bevacizumab are the only mAb 46 based drugs that are approved for the treatment of patients with metastatic CRC [5-7].”. At present immune checkpoint inhibitors have been approved for the first-line treatment of patients with unresectable or metastatic microsatellite instability-high (MSI-H) or mismatch repair deficient (dMMR) colorectal cancer. In particular, pembrolizumab, a mAb against PD-1, is used for the treatment of mCRC. Revise the sentence accordingly. Fort this purpose, see:
- https://www.fda.gov/news-events/press-announcements/fda-approves-first-line-immunotherapy-patients-msi-hdmmr-metastatic-colorectal-cancer
- 10.3390/cancers11101472
- 10.1177/1756284820917527
- 10.15171/bi.2019.16
4) In the Results section, the authors should present data about PFS and OS of patients with co-expression of two, three and four HER family members. Have the authors performed such analyses? For example, in Table 2 they showed that 56 patients had the co-expression of HER2, HER3, and HER4. Are there statistical differences in Kaplan-Meier curves of these 56 patients compared to patients with the only expression of HER2, HER3 or HER4? Please, provide these data to propose a prognostic score based on the co-expression of HER family members;
5) In the following sentence, the authors provided a reference for necitumumab but not for cetuximab and panitumumab: “The approval of the anti-EGFR mAbs, cetuximab and panitumumab for the treatment of mCRC and in recent years, necitumumab for lung cancer [24], emphasises the importance of EGFR as a therapeutic target in human cancers.”. Please, provide references supporting these notions. For this purpose, see:
- 10.1097/MJT.0b013e31826a94d8
- 10.1634/theoncologist.12-5-577
- 10.3389/fphar.2018.01300
6) In line 231-236, please specify that the authors are referred to the expression of the wild type form of EGFR;
7) In the Discussion section, the authors should mention the existing clinical trials assessing the efficacy of trastuzumab or other HER family inhibitors. For this purpose, please check if clinical trials are recorded on EudraCT or clinicaltrial.gov repositories.
Author Response
Reviewer 1 Comments
1) As the authors use the name of the active principles cetuximab and panitumumab in the Abstract section, they should be consistent using the name “trastuzumab” instead of the commercial one “Herceptin”;
Our response: Thank you. Done.
2) In line 40-42, please avoid the repetition of “in the USA”;
Our response: Thank you. The manuscript has been revised and repetition is removed.
3) Please, revise the following sentence: “Currently, the anti-epidermal growth factor receptor (EGFR) monoclonal antibodies (mAbs) cetuximab and panitumumab and 45 anti-vascular endothelial growth factor (VEGF) mAb bevacizumab are the only mAb 46 based drugs that are approved for the treatment of patients with metastatic CRC [5-7].”. At present immune checkpoint inhibitors have been approved for the first-line treatment of patients with unresectable or metastatic microsatellite instability-high (MSI-H) or mismatch repair deficient (dMMR) colorectal cancer. In particular, pembrolizumab, a mAb against PD-1, is used for the treatment of mCRC. Revise the sentence accordingly. Fort this purpose, see:
- https://www.fda.gov/news-events/press-announcements/fda-approves-first-line-immunotherapy-patients-msi-hdmmr-metastatic-colorectal-cancer
- 10.3390/cancers11101472
- 10.1177/1756284820917527
- 10.15171/bi.2019.16
Our response: Thank you. We have revised the manuscript and included details of all the other approved antibodies including pembrolizumab with the supporting references.
4) In the Results section, the authors should present data about PFS and OS of patients with co-expression of two, three and four HER family members. Have the authors performed such analyses? For example, in Table 2 they showed that 56 patients had the co-expression of HER2, HER3, and HER4. Are there statistical differences in Kaplan-Meier curves of these 56 patients compared to patients with the only expression of HER2, HER3 or HER4? Please, provide these data to propose a prognostic score based on the co-expression of HER family members.
Our response: Thank you for raising this important point. Indeed, we had performed statistical analysis for the various co-expressions of the HER family differences in both PFS and OS. However, we found no statistically significant differences. We have now added a sentence in the revised manuscript to highlight this.
5) In the following sentence, the authors provided a reference for necitumumab but not for cetuximab and panitumumab: “The approval of the anti-EGFR mAbs, cetuximab and panitumumab for the treatment of mCRC and in recent years, necitumumab for lung cancer [24], emphasises the importance of EGFR as a therapeutic target in human cancers.”. Please, provide references supporting these notions. For this purpose, see:
- 10.1097/MJT.0b013e31826a94d8
- 10.1634/theoncologist.12-5-577
- 10.3389/fphar.2018.01300
Our response: Thank you for this comment. We have provided supported references for cetuximab (5) and panitumumab (7) earlier in introduction. Therefore, we revised the manuscript by adding these references again and other supporting references.
6) In line 231-236, please specify that the authors are referred to the expression of the wild type form of EGFR;
Our response: Thank you. Done.
7) In the Discussion section, the authors should mention the existing clinical trials assessing the efficacy of trastuzumab or other HER family inhibitors. For this purpose, please check if clinical trials are recorded on EudraCT or clinicaltrial.gov repositories.
Our response: Thank you. While we have already mentioned findings of published clinical studies that assess the efficacy of trastuzumab and other HER family inhibitors in the relevant part of the discussion section (References: 41, 43-45 & 46), we have added examples of some relevant ongoing clinical trials.
Reviewer 2 Report
In their manuscript to Cancers Khelwatty et. all present quantifications of immunohistochemical staining of colorectal cancer samples and suggest that HER2 expression is predictive of patient survival upon Cetuximab-treatment of wild-type Ras expressing tumours.
Criticism:
1.Authors used 4 different antibodies (EGFR, HER2, HER3, HER4 -one ab against each) and state that the antibodies were specific. This is not sufficient, especially when only one antibody per antigen was used. Authors should provide proofs of specificity for these antibodies.
2.Authors specifically study Ras wt tumors, but do not state anywhere how the Rat status was detected, which method was used. Was the Rat mutation status detected before or after the treatment?
3. Authors should present in the supplementary material examples of the staining with different antibodies for the different scorings 0,1+,2+,3+
4.At lane 131-132 "from mCRC patients before and after treatment with wild-type RAS". Were the patients or the tissue samples treated with WT-Ras and why?
5.Authors should discuss why others e.g. in PMID 28223103 do not get similar results.
Author Response
Reviewer 2 Comments
1.Authors used 4 different antibodies (EGFR, HER2, HER3, HER4 -one ab against each) and state that the antibodies were specific. This is not sufficient, especially when only one antibody per antigen was used. Authors should provide proofs of specificity for these antibodies.
Our response: We thank the reviewer for raising this important point. However, we would like to highlight that these antibodies were all commercially sourced (details of the antibody & supplier provided in the manuscript) and optimised using known positive samples and/or tissues as in our previously published studies (Reference 21, 26).
- Authors specifically study Ras wt tumors, but do not state anywhere how the Rat status was detected, which method was used. Was the Rat mutation status detected before or after the treatment?
Our response: Thank you. The manuscript has been revised and further information relating the RAS status has been added.
- Authors should present in the supplementary material examples of the staining with different antibodies for the different scorings 0,1+,2+,3+
Our response: Thank you. We have now included the staining with different antibodies and scorings as supplementary figure as suggested.
- At lane 131-132 "from mCRC patients before and after treatment with wild-type RAS". Were the patients or the tissue samples treated with WT-Ras and why?
Our response: Thank you for highlighting this typographical error which has been revised.
5.Authors should discuss why others e.g. in PMID 28223103 do not get similar results.
Our response: We thank the reviewer for highlighting this study, however it is noteworthy that the study concludes “HER2 amplification is predictive of shorter PFS after cetuximab treatment in patients with mCRC harbouring wild-type RAS and BRAF”, which is largely in concordance with the findings of our study. Indeed there are some vital methodological differences in determining the expression of HER2 in the above study for example, a major difference being the use of a different primary antibody against HER2 (4B5) and use of tissue microarrays which could have contributed to a lower detectable expression of HER2 compared to our study.
Round 2
Reviewer 1 Report
The authors well addressed all of my previous comments. There are no further issues to clarify.
Author Response
We thank the reviewer 1 for their constructive comments.
Reviewer 2 Report
I have few complains left.
-First of all authors still don't explain how the Ras status was determined, which is the important criteria for their sample selection.Who did it, where and which method. I cannot find this information from the manuscript. Authors thought state in their letter that they did. It should be mentioned in the methods.
-There is a sentence (lines 147-148) added:
"It is also noteworthy that in this study we did not find the co-expression of HER family members to have a significant statistical difference in both OS and PFS survival analyses (Data not shown). " The data should be shown, since it exists, in supplementary figures, if authors want to publish this information.
